# Effects of Filtration Mode on the Performance of Gravity-Driven Membrane (GDM) Filtration: Cross-Flow Filtration and Dead-End Filtration

**Qian Wang** [1], **Xiaobin Tang** [1], **Heng Liang** [1], **Wenjun Cheng** [1], **Guibai Li** [1], **Qingjun Zhang** [2], **Jie Chen** [3], **Kang Chen** [3] and **Jinlong Wang** [1,*]

1 State Key Laboratory of Urban Water Resource and Environment, School of Environment, Harbin Institute of Technology, Harbin 150090, China; fswaterhit@163.com (Q.W.); tang5462@163.com (X.T.); hitliangheng@163.com (H.L.); chengwenjun178@163.com (W.C.); liguibai@vip.163.com (G.L.)

2 Taizhou Construction Industry Development Affairs Center, Taizhou 318001, China; qingjunzhang2010@163.com

3 Nollet Intelligent Water Equipment Co., Ltd., Suzhou 226334, China; hitchenjie@163.com (J.C.); szchenkang@163.com (K.C.)

* Correspondence: chnwangjinlong@163.com; Tel.: +86-157-6553-2017

**Abstract:** Gravity-driven membrane (GDM) filtration technology has been extensively in the employed drinking water treatment, however, the effect filtration mode (i.e., dead-end mode vs. cross-flow mode) on its long-term performance has not been systematically investigated. In this study, pilot-scale GDM systems were operated using two submerged filtration mode (SGDM) and cross-flow mode (CGDM) at the gravity-driven pressures 120 mbar and 200 mbar, respectively. The results showed that flux stabilization was observed both in the SGDM and CGDM during long-term filtration, and importantly the stabilized flux level of CGDM was elevated by 3.5–67.5%, which indicated that the filtration mode would not influence the occurrence of flux stability, but significantly improve the stable flux level. Interestingly, the stable flux level was not significantly improved with the increase of driven pressure, and the optimized driven pressure was 120 mbar. In addition, the GDM process conferred effective removals of turbidity, $UV_{254}$, $COD_{Mn}$, and DOC, with average removals of 99%, 43%, 41%, and 20%, respectively. With the assistance of cross flow to avert the overaccumulation of contaminants on the membrane surface, CGDM process exhibited even higher removal efficiency than SGDM process. Furthermore, it can be found that the CGDM system can effectively remove the fluorescent protein-like substances, and the intensities of tryptophans substance and soluble microbial products were reduced by 64.61% and 55.08%, respectively, higher than that of the SGDM. Therefore, it can be determined that the filtration mode played an important role in the flux stabilization of GDM system during long-term filtration, and the cross-flow filtration mode can simultaneously improve the stabilized flux level and removal performance.

**Keywords:** gravity-driven membrane (GDM); flux stabilization; ultrafiltration; dead-end filtration; cross-flow filtration; membrane fouling

## 1. Introduction

The gravity-driven membrane (GDM) ultrafiltration has been increasingly reported in the fields of surface water treatment, rainwater reclamation, and seawater pretreatment [1,2]. In GDM filtration, ultra-low transmembrane pressure (TMP, 20~70 mbar) was adopted, and the membrane system can be performed continuously at a stable flux level (4–7 L m$^{-2}$ h$^{-1}$) without any hydraulic backwashing and chemical cleaning for the specific requirements of membrane fouling control by [3]. As a consequence, the GDM technology has the merits of chemical-free addition, low energy consumption, simple operation, and low maintenance, and has been regarded as a sustainable and cost-effective technology for decentralized water purification.

During the GDM filtration, colloids, particles, and microorganisms, would be effectively intercepted by the ultrafiltration (UF) membrane, and thereby aggregated on the membrane surface [4]. Consequently, these increasingly accumulated substances would gradually develop a fouling layer on the UF membrane surface, which is determined as a "so-called" biofouling layer [5]. In such situations, the GDM system can couple the multi-functional rejection between the biofouling layer and UF membrane, enhancing the removal performance of organic pollutants (i.e., biopolymers and assimilable organic matter (AOC)) and pathogens in the feed water [6].

In addition, the formation of the biofouling layer on the UF membrane surface was reported to be closely related to the flux stabilization of GDM process, especially its structural and compositional characteristics. Derlon et al. indicated that improving the roughness and porosity of the biofouling layer can spark 100% improvement of stable flux by adding protozoa (e.g., *Tetrahymena pyriformis*) to the GDM system to enhance the biological predation within the biofouling layer [5,7]. Conversely, when the biological activity was suppressed, the structures of biofouling layer exhibited dense and compact, and consequently the GDM permeation decreased significantly.

In order to improve the removal performance and permeability of GDM process, some simple approaches have been developed. Chomiak et al. [6] adopted the slow filter and biological filter to pre-remove pollutants from the feed water before the GDM unit, contributing to stable permeation of GDM increasing by about 29% and 12%, respectively. Shao et al. [8] coated granular media (PAC, mesoporous adsorption resin and silica sand) on the membrane surface of GDM, and thus significantly improved the removal rate of organic matters (such as algal toxin, bisphenol A, and atrazine). However, the stable permeation of the GDM process became lower than the control. Wu et al. [9] indicated that the packing density of the membrane module dominated the living space of the predators in the biofouling layer, which would control the stable permeation and removal efficiency. Akhondi et al. [10] used GDM to treat seawater and found that when the TMP increased from 40 mbar to 100 mbar, the stable permeation increased by $1 \sim 1.6$ L m$^{-2}$ h$^{-1}$; thus, the TMP effectively controlled the GDM. Peter-Varbanets et al. [11] tested different TMP (40, 150, 250, and 500 mbar) to execute the GDM process. The experiments showed that the level of flux stabilization did not depend on pressure. Regarding the effect of TMP on the GDM operation, there exist conflicting points of view in the scientific community.

The filtration modes, including dead-end filtration and cross-flow filtration, play an important role in controlling the conventional UF membrane flux. In conventional UF filtration, compared to dead-end filtration, cross-flow filtration can obtain better flow distribution and effectively reduce concentration polarization [12]. Compared with cross-flow filtration, dead-end filtration can promote foulants mainly depositing the membrane inside, which makes it easy to clean out the fouling [13]. Besides, dead-end filtration also has the advantages of low operating pressure, low energy consumption, small footprint, and low operating cost [14].

In GDM filtration, Shi et al. [15] found that the combination of shear stress and membrane relaxation promotes the biofouling layer loose and heterogeneous, and contributes to the formation of cavities and channel network structure in this layer, enhancing the stable permeability of GDM by 70%. Nevertheless, Ding et al. [16] analyzed the effect of aeration shear stress on the GDM system used for gray water treatment. When the aeration was placed below the membrane module, the shear stress caused the biofouling layer to be thinner and denser [17]. The stable permeability was lower than that without aeration shear. To the best of our knowledge, few reports about the effects of filtration mode on the GDM were observed, and the pilot-scale GDM investigations on different filtration modes have not been reported so far.

In this study, four pilot-scale GDM systems were introduced to reveal the effects of the filtration mode (e.g., dead-end filtration vs. cross-flow filtration) on the flux stabilization and the flux levels were analyzed. The removal performance of different contaminants was also systematically investigated. The influence of different driven pressures on the GDM long-term filtration was estimated. This study is expected to provide new insights to the performance improvements and practical application of GDM technology.

## 2. Materials and Methods

### 2.1. Raw Water

In this experiment, the raw water was pumped from a reservoir in Wenchang city, Hainan province of China. This reservoir has typical characteristics of water quality, representing many drinking-water sources globally. The reservoir contains high levels algae in summer. Moreover, the turbidity of feed water was easily affected by the rainstorm. The turbidity can significantly increase to more than around 10 NTU when continuous rain. The raw water qualities are summarized in Table 1.

**Table 1.** Characteristics of raw water quality during the experiment.

| Characteristics | Values |
|:---:|:---:|
| $COD_{Mn}$ | 1~3 mg $L^{-1}$ |
| $UV_{254}$ | 0.018~0.045 $cm^{-1}$ |
| Ammonia | 0.07~0.15 mg $L^{-1}$ |
| Turbidity | 1~6.5 NTU |
| Dissolved oxygen | 4~7 mg $L^{-1}$ |
| pH | 6.8~7.6 |
| Temperature | 16~33 °C |

### 2.2. Experimental Setups and Operation Procedures

Two pilot-scale setups, including the submerged gravity-driven membrane filtration system (SGDM, Figure 1a) and cross-flow gravity-driven membrane filtration system (CGDM, Figure 1b), were made and continuously operated during the entire experiment period. The hollow-fiber UF membranes were commercial with an average pore size of 0.1 μm, the inner diameter of 1.0 mm, and the outer diameter of 1.6 mm, respectively. Each UF module contained 10 $m^2$ membranes.

The operation procedures of the two systems were as follows. The raw water flowed into a raw water tank where it had a constant water level. Then, the water directly entered the SGDM or CGDM. The outlet of GDM permeate was immersed in an overflowed permeate tank where it also had a constant water level. These two constant water levels supplied the gravity-driven pressure for GDMs. The GDM permeates were sampled from the permeate tank for characterization.

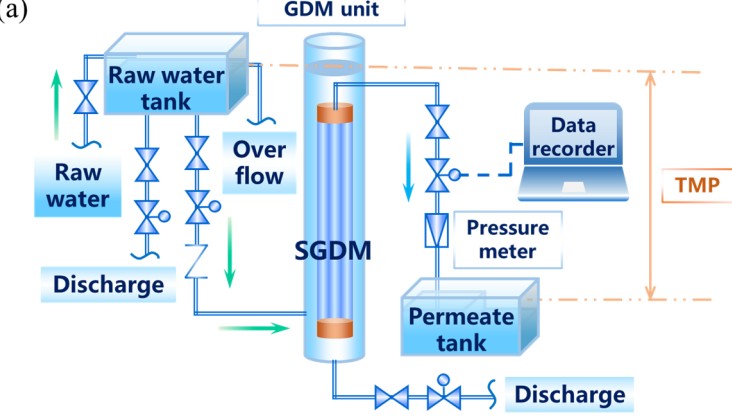

(a)

**Figure 1.** *Cont.*

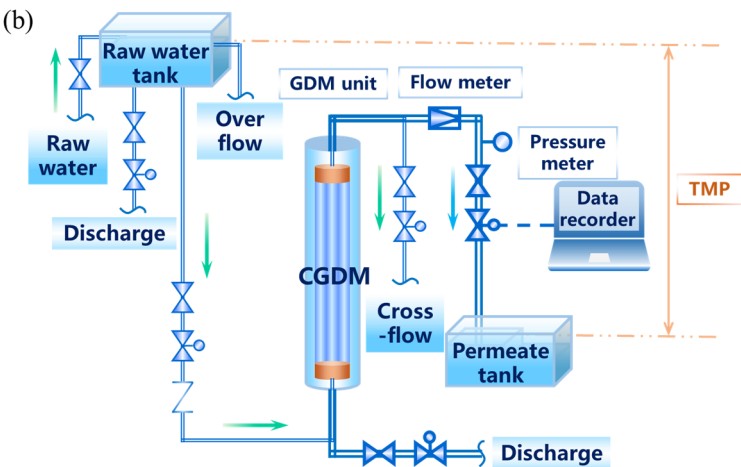

**Figure 1.** Schematic diagram of the experimental setups: (**a**) SGDM, (**b**) CGDM.

In this study, GDMs were applied without any pretreatment, hydraulic backwashing, and chemical cleaning procedures. The only different procedure between the SGDM and CGDM was that the raw water entered the UF module with cross-flow mode and flowed out after a flow valve. The cross-flow flux was regulated with 22.5 L m$^{-2}$ h$^{-1}$ by the flow value to reveal the effect of cross-flow mode on the GDM performance. The detailed information of SGDM and CGDM was compared in Table 2.

**Table 2.** Comparison of SGDM and CGDM systems.

| Systems | Group Name | TMP (mbar) |
|---------|------------|------------|
| SGDM | SGDM-120 | 120 |
|  | SGDM-200 | 200 |
| CGDM | CGDM-120 | 120 |
|  | CGDM-200 | 200 |

*2.3. Analytical Methods*

The permanganate index (COD$_{Mn}$), ammonia nitrogen (NH$_4^+$-N) and the total number of bacteria used in this experiment were determined according to the Chinese National Standard (GB/T 5750.4-2006). The turbidity was measured with a turbidity meter (2100Q portable turbidity meter, Hatch®, Danaher Corporation, New York, NY, USA). The concentration of dissolved organic compounds (DOC) was measured with a TOC analyzer (MultiN/C2100S, Analytic Jena®, Endress+Hauser Group, Jena, Thuringia, Germany) after pre-filtration of samples using 0.45 μm filters. Organic matter was also determined by ultraviolet-visible spectrophotometry (UV752N, Jingke®, Shanghai Precision Science Instrument Limited Company, Shanghai, China) at a wavelength of 254 nm (UV$_{254}$). Fluorescence excitation-emission spectrometry (F7000, Hitachi®, Hitachi Company, Tokyo, Japan) was used to measure fluorescent organic compounds. The excitation spectrum and emission spectrum were scanned from 200 nm to 450 nm in 5 nm increments, and 250 nm to 550 nm in 1 nm increments, respectively. The dissolved oxygen was measured with a dissolved oxygen meter (Multi 3420, WTW®, Xylem Analytics Germany Sales GmbH & Co. KG, Weilheim, Bavaria, Germany). Membrane fouling resistances were calculated according to a previous study [18].

**3. Results and Discussion**

*3.1. Membrane Permeation and Membrane Resistance*

Membrane permeation and membrane resistance are important ways of characterizing membrane fouling. The SGDM systems were run with 120 mbar and 200 mbar

respectively as the filtration heads. The membranes were cleaned physically and chemically before this study. As shown in Figure 2a, the initial membrane penetrations of the SGDM-120 and the SGDM-200 were 16.75 L m$^{-2}$ h$^{-1}$ and 29.40 L m$^{-2}$ h$^{-1}$, respectively. After 23 days of operation, these two membrane permeations both reached a stable state, and the average permeations were 6.68 L m$^{-2}$ h$^{-1}$ and 10.78 L m$^{-2}$ h$^{-1}$, respectively. As for the CGDMs, the initial membrane penetrations in the CGDM-120 and CGDM-200 were 29.56 L m$^{-2}$ h$^{-1}$ and 32.94 L m$^{-2}$ h$^{-1}$, respectively. After the 27th day, the membrane permeations reached a stable level. The fluxes at the end of the experiment were 11.19 L m$^{-2}$ h$^{-1}$ and 10.42 L m$^{-2}$ h$^{-1}$. Although the initial permeations of different GDM systems were significantly different, the stable permeations tended to be a similar level. From the comparing difference of UF permeations between four GDMs, it can be found that the initial permeations of the two systems of CGDM were obviously higher than that of the two of the SGDM system. This result may be related to the cross-flow mode applied in the CGDM systems, which reduced the deposition of foulants onto the membrane surface and the blockage of the membrane pores [15].

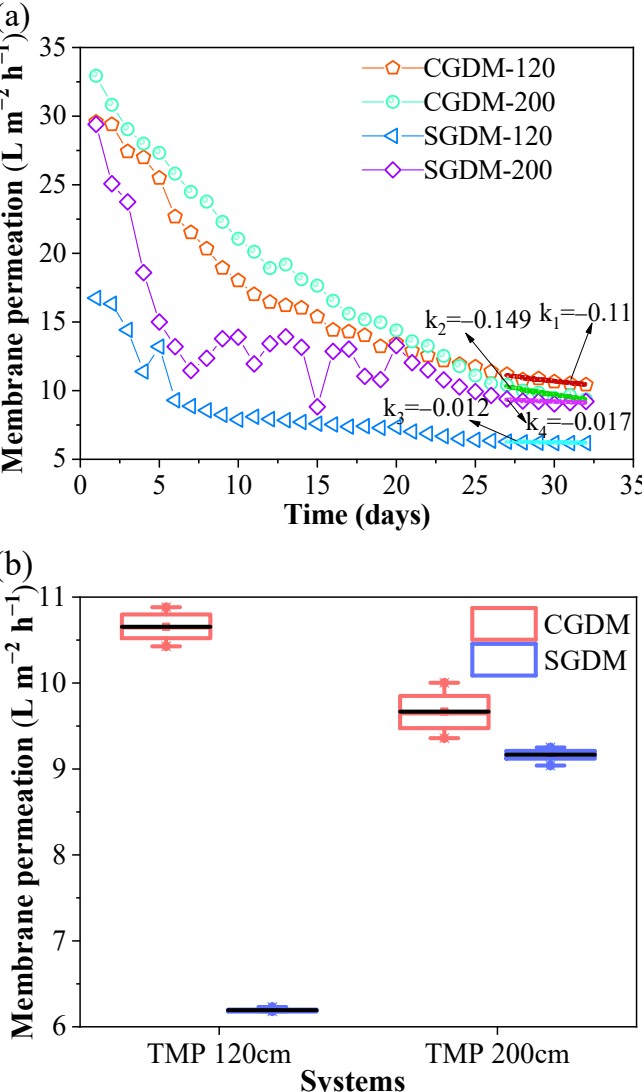

**Figure 2.** Trends of membrane permeation and membrane resistance: (**a**) membrane permeation, (**b**) stable membrane permeation (28–32 days).

The permeations of the CGDMs were delayed compared to the SGDM system. The reason might be that the cross-flow mode controlled the CGDM systems' influent for

a long time in a turbulent state, causing living microorganisms difficult to adhere to the surface of the UF membrane and thereby hard to form the critical biofouling layer. Comparing the permeations after 28 days of filtration (Figure 2b), it is found that the final stable permeations of the CGDM systems were significantly higher than that of the SGDM systems. Besides, the stable permeation of CGDM-120 was much higher than SGDM-120. Thus, the lower TMP contributed to the superiority of cross-flow GDM, which had a higher stable membrane permeation than the submerged GDM. This result proved the feasibility of cross-flow mode for obtaining high UF flux during GDM filtration.

### 3.2. Removal Performance of Various Pollutants

3.2.1. Turbidity

As shown in Figure 3a, the of in GDMs purified the effluents with quietly low turbidity (<0.1 NTU) due to the barrier role of UF membranes [19]. Moreover, the GDM system maintained stable turbidity removal performance and membrane permeation while the raw water turbidity suddenly varied. It was shown that GDMs had a high buffer capacity in turbidity removal, consistent with previous study [20]. For all GDM systems, the removal rate of turbidity was higher than 99% after filtration of 15 days (Figure 3b).

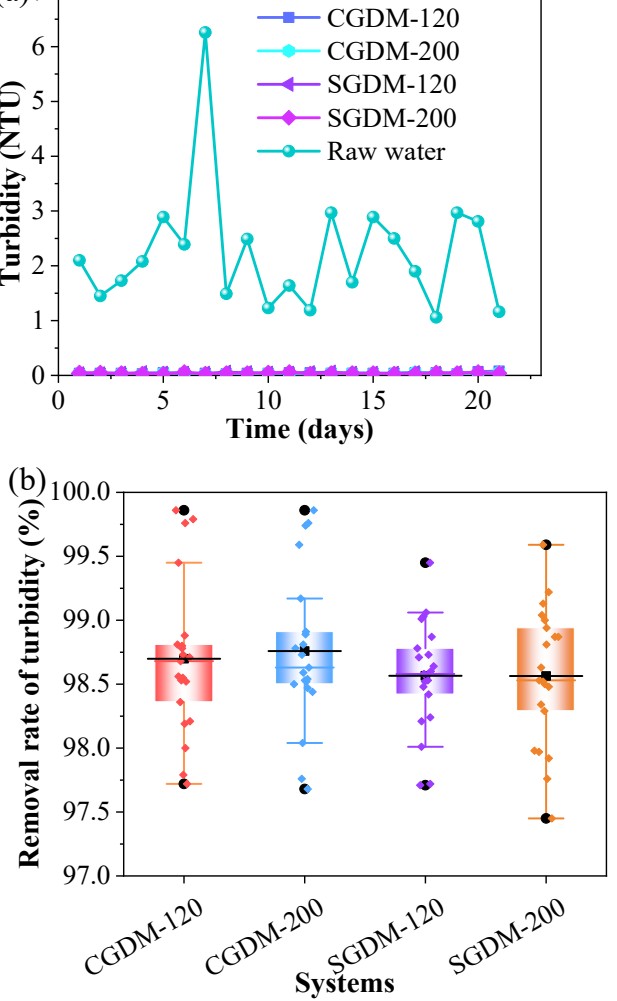

**Figure 3.** Removal performance of turbidity: (**a**) turbidity in effluents, (**b**) removal rate of turbidity (including average, maximum, and minimum).

### 3.2.2. Organic Matters

As shown in Figure 4a, during the entire filtration, despite a large change to raw water quality, GDMs still purified the effluent with a stable $UV_{254}$ index value. In Figure 4b, the $UV_{254}$ in the effluent of the SGDM was higher than that of the CGDM, especially after the initial 10 days of filtration. In the dead-end GDM systems, natural organic matters (NOM) were converted to matters with low UV absorption or permeate out. However, in the cross-flow GDMs, some NOM could be discharged from the cross-flow pipeline, reducing the total mass of NOM on the membrane surface. The results indicated CGDMs removed organic matter of $UV_{254}$ better than the SGDMs.

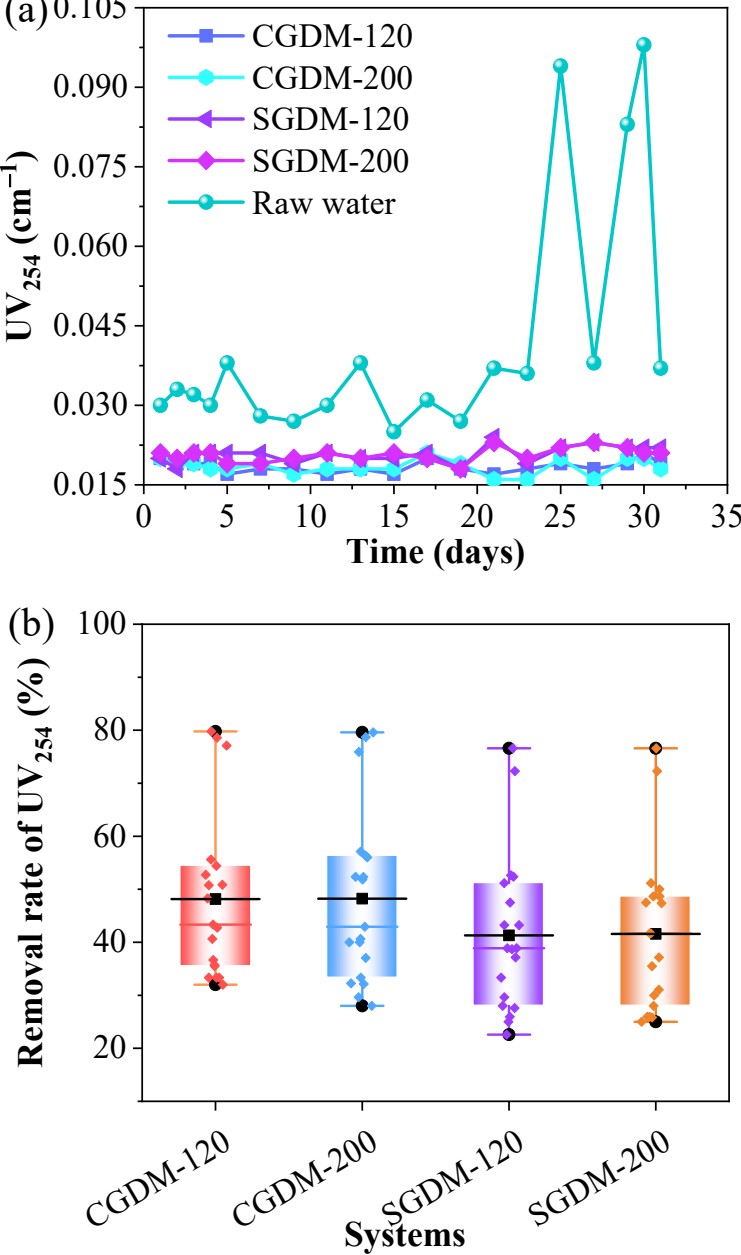

**Figure 4.** *Cont.*

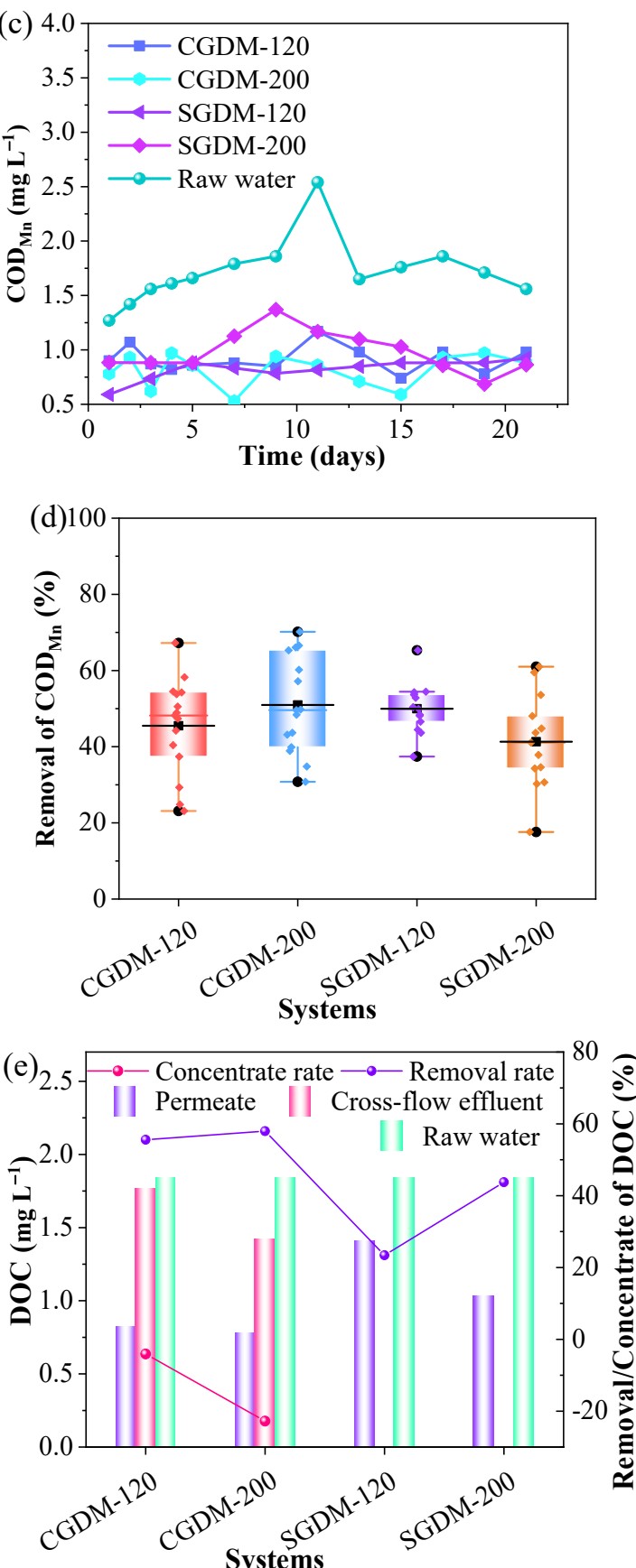

**Figure 4.** Removal performance of organic matters: (**a**) trend of $UV_{254}$, (**b**) removal rate of $UV_{254}$, (**c**) trend of $COD_{Mn}$, (**d**) removal rate of $COD_{Mn}$, (**e**) removal performance of DOC.

As shown in Figure 4c,d, the $COD_{Mn}$ concentration of CGDM effluent was varied between 0.5 mg $L^{-1}$ and 1.3 mg $L^{-1}$, while the SGDM system showed the $COD_{Mn}$ concentration with a narrow range. From the perspective of TMP, the gravity-driven pressure had a slight effect on the removal rate of $COD_{Mn}$ in the CGDMs, but it had a significant impact on the SGDMs. Nevertheless, the removal rates were generally between 40% and 60% in all GDM systems. As for $COD_{Mn}$ concentration, the CGDM produced more stable drinking water than the SGDM when their TMP frequently changed.

According to Figure 4e, the DOC concentrations of the raw water were around 1.80 mg $L^{-1}$, while its concentrations of effluents were lower than 1.50 mg $L^{-1}$. The results implied that the microorganisms transformed more organic matters to inorganic matters rather than hydrolyze them, reducing the DOC concentration in the UF permeate. In the CGDM systems, the average DOC concentrations in the membrane permeate were 0.817 mg $L^{-1}$ (with a removal rate of 55.58%) and 0.774 mg $L^{-1}$ (with a removal rate of 57.95%), respectively. As for the SGDM systems, the average DOC concentrations of the effluent were 1.409 mg $L^{-1}$ in the SGDM-120 system and 1.035 mg $L^{-1}$ in the SGDM-200 system, and the removal rates were 23.42% and 43.75%, respectively. Compared with the SGDM, the cross-flow mode improved the DOC removal rate of GDM by 10–30%. As for the two CGDM systems, the DOC concentrations in UF membrane permeates were slightly effect by the TMP variation. In the CGDM, the biofouling layer still formed despite the flush behavior of the cross flow. Besides, the cross-flow filtration mode, cross-flow intensity, and cross-flow fluid state together controlled the DOC-contained substances far away from the membrane surface. Overall based on the DOC concentration, the CGDM was better than the SGDM in the removal rate and removal stability.

### 3.2.3. Trend of DO Concentration

During the experiment, the surface water of the reservoir was used for filtration and the DO concentration in the raw water was between 3 and 7 mg $L^{-1}$. As shown in Figure 5, prolonging filtration time of the CGDM, the DO decrease in the membrane column was continuously increased from 15.14% to 31.13%, indicating that the consumption of DO by the biomass in the CGDM. After the 30th day, the DO consumption of the CGDM system was slightly decreased, but the average decrease rate of DO in these systems still exhibited above 20%. The SGDM system showed a similar trend to the CGDM, but the SGDM system obviously consummated a higher concentration of DO than the CGDM system. It might be caused by the higher biomass of the biofouling layer, and thereby the microorganism consumed more oxygen.

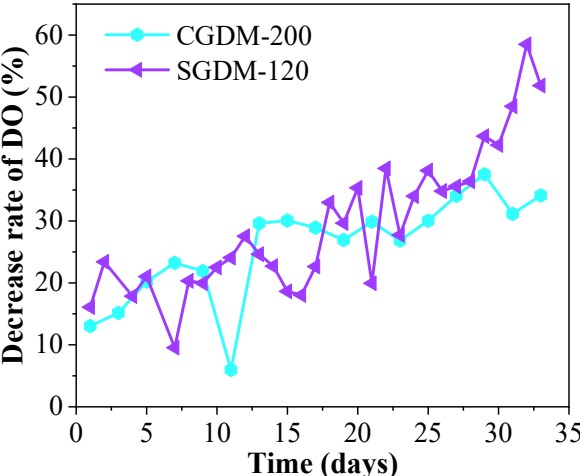

**Figure 5.** Trend of DO concentration.

### 3.2.4. Fluorescent Excitation-Emission Matrix

The three-dimensional fluorescence spectra of raw water, SGDM permeate, and CGDM permeate are shown in Figure 6. Table 3 lists their fluorescence positions and fluorescence intensities. According to fluorescence spectrum results, four types of fluorescent substances were detected in raw water. Peak A, peak B, peak C, and peak D respectively represented protein peaks (tyrosine), protein peaks (tryptophans), humus, and soluble microbial products (SMP) [21–23]. The raw water contained above four peaks, while the permeate from the SGDM and CGDM systems only showed two main peaks, including peak B and peak C. The results indicated that the GDM systems effectively can remove tyrosine proteins (a fluorescent protein-like substance) and SMP substances. As seen from Table 3, the intensities of peaks A, B, C, and D in the raw water were 273.27 a.u., 333.19 a.u., 149.84 a.u., and 198.40 a.u.. The intensities of peak B and peak C in the SGDM effluent were 145.12 a.u. and 106.94 a.u., respectively, reaching the peak intensity reduction rate of 56.45% and 28.63%. For the CGDM, the intensities of peaks B and C in its effluent were 117.92 a.u. and 89.13 a.u., and the peak intensity reduction rates were about 64.61% and 55.08%, respectively. It can be found that the CGDM system had better ability than the SGDM system in removing tyrosine proteins and SMP. These results were consistent with the DOC concentrations. From the above figure and table, it can also be found that CGDM and SGDM selectively removed different fluorescent organics, especially had the advantage of removing tyrosine-based protein substances and SMP substances but had weak removal of tryptophan-based proteins and humic substances. In addition, compared to the traditional GDM system (SGDM), the CGDM system had a better removal effect on soluble microbial products.

### 3.3. Comparison for Applications of Gravity-Driven Membrane Filtration

GDM has been widely used in drinking water treatment, as shown in Table 4. In the previous literature, the TMP of GDM was controlled between 40 and 150 mbar with a stable membrane flux of 3.6–10.6 L m$^{-2}$ h$^{-1}$. However, these studies mainly focused on the lab-scale membrane flux and its mechanism rather than removal performances. Nicolas et al. [24] studied the GDM processes and compared the two filtration modes including cross-flow mode and dead-end mode. It was found that the cross-flow mode increased the stable membrane flux for GDM, consistent with the results of this study. Ding et al. [25] and Wu et al. [9] used GDM to treat rainwater and seawater, respectively. The results showed that the DOC removal approached zero for the GDM and even the DOC concentration in membrane permeate was higher than the feedwater. This might be caused by the more active bio-hydrolysis than the degradation of microorganisms. Song et al. [26] treated reservoir water using GDM, the removal rate exhibited a high level of 63.8%, which agreed with this study. This may be related to the content of assimilable organic carbon in the reservoir water being higher than the rainwater and seawater. Thus, the intensity of biodegradation was stronger than the intensity of bio-hydrolysis. In general, based on this work and previous research, cross-flow can play a positive role in GDM.

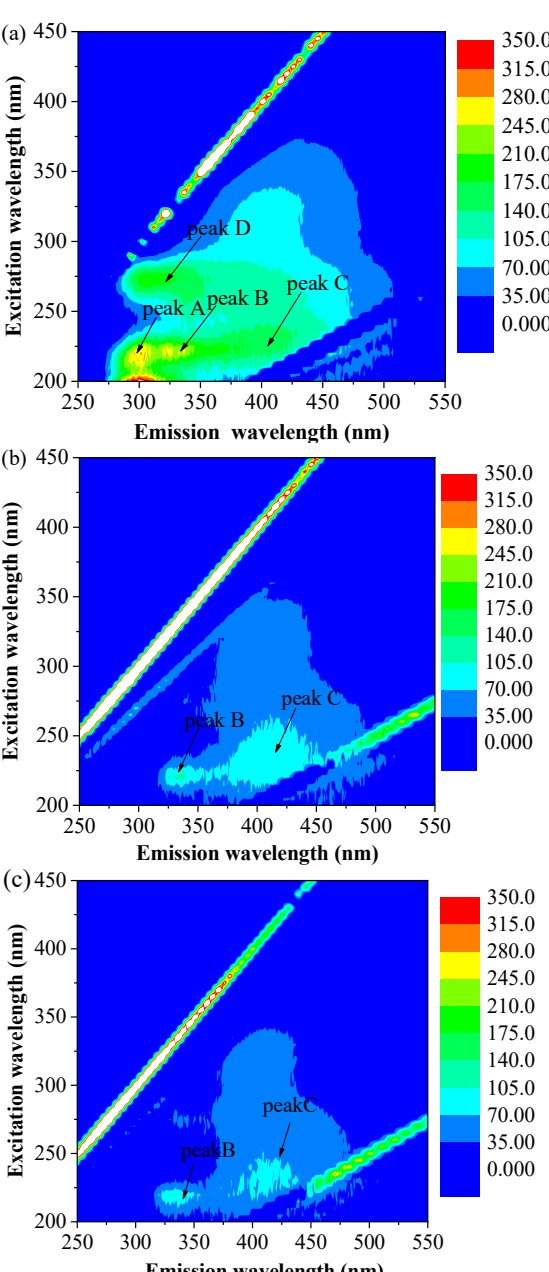

**Figure 6.** The effects of filtration modes on fluorescent excitation-emission matrix (EEM) in the water samples: (**a**) raw water, (**b**) average results in the permeates of two SGDMs, (**c**) average results in the permeates of two CGDMs.

**Table 3.** Effects of filtration modes on fluorescent peak intensity of EEM.

| Water Samples | Peak A | | Peak B | | Peak C | | Peak D | |
|---|---|---|---|---|---|---|---|---|
| | Ex/Em (nm/nm) | Intensity (a.u.) | Ex/Em (nm/nm) | Intensity (a.u.) | Ex/Em (nm/nm) | Intensity (a.u.) | Ex/Em (nm/nm) | Intensity (a.u.) |
| Raw water | 220/303 | 273.27 | 220/326 | 333.19 | 230/410 | 149.84 | 275/306 | 198.40 |
| SGDM | — | — | 220/330 | 145.12 | 230/410 | 106.94 | — | — |
| IGDM | — | — | 220/329 | 117.92 | 230/408 | 89.13 | — | — |

**Table 4.** Comparison for applications of GDM filtration.

| Water Source | Flux (L m$^{-2}$ h$^{-1}$) | Rejection Performance (%) | | TMP (mbar) | Filtration Mode * | Membrane Type ** | Ref. |
|---|---|---|---|---|---|---|---|
| | | COD$_{Mn}$ | DOC | | | | |
| Reservoir water | 10.65 | 45.5 | 55.6 | 120 | CM | HF, PVDF (0.1 μm) | This work |
| Reservoir water | 9.16 | 30.6 | 23.4 | 200 | DM | HF, PVDF (0.1 μm) | This work |
| River water | 8.0 | – | – | 40 | DM | FS, PES (100 kDa) | [7] |
| Rain water | 6.0 | – | towards zero | 50 | DM | FS, PES (150 kDa) | [25] |
| Seawater treated by biofilm reactor | 4.5 | – | 3.6 | 40 | DM | FS, PES (100 kDa) | [9] |
| Seawater treated by biofilm reactor | 3.6 | – | –29.7 | 40 | DM | HF, PVDF (0.1 μm) | [9] |
| Reservoir water | 4.0 | 80.2 | 63.8 | 150 | DM | HF, PVDF (0.08 μm) | [26] |
| River water | 10.9 | – | – | 100 | CM | FS, PES (150 kDa) | [24] |
| River water | 6.7 | – | – | 100 | DM | FS, PES (150 kDa) | [24] |

Notes: * Filtration mode: CM (cross-flow mode), DM (dead-end mode). ** Membrane type: HF (hollow fiber), FS (flat sheet), PVDF (poly (vinylidene fluoride)), PES (polyethersulfone).

## 4. Conclusions

In this study, the effects of filtration mode on GDM long-term performance were investigated, and the influence of driven pressure (120 mbar and 200 mbar) was also estimated. The following conclusions can be drawn:

(1) During long-term filtration, different filtration modes (dead-end filtration and cross-flow filtration) would not impact the occurrence of flux stabilization, but lead to different stabilized flux levels. The stable flux of CGDM was obviously higher than that of the SGDM system at the same driven pressure condition, since cross-flow operation could prevent the overaccumulation of contaminants on the membrane surface.

(2) GDM systems had effective removal ability of turbidity, achieving a removal rate of 99% after filtration of 15 days. Besides, similar removal efficiencies were achieved both in SGDM and CGDM, which indicated that the filtration mode would not influence the rejection of turbidity.

(3) The average removal efficiencies of UV$_{254}$, COD$_{Mn}$, and DOC in CGDM were 48.1%, 48.2%, and 56.8%, slightly higher than those in CGDM since the cross-flow operation would avoid the accumulation of organic foulants on the membrane surface to reduce their hydrolysis into the membrane effluent.

(4) During long-term filtration, the DO concentration decreased gradually and finally came to a decrease rete of approximately 20% after 30 days of filtration, due to DO consumption by the biological activity within the biofouling layer. Besides, the SGDM system exhibited higher consumption rate of DO concentration than the CGDM.

(5) The CGDM system could effectively remove the fluorescent organic compounds (especially the protein-like substances), with the intensities of tryptophans and soluble microbial products being reduced by 64.61% and 55.08%, respectively, higher than the SGDM system.

**Author Contributions:** Conceptualization, J.W. and K.C.; methodology, J.C.; software, Q.W.; validation, X.T.; formal analysis, Q.W.; investigation, Q.W.; resources, J.W.; data curation, W.C. and Q.Z.; writing—original draft preparation, Q.W.; writing—review and editing, Q.W. and J.W.; visualization, W.C.; supervision, G.L. and Q.Z.; project administration, H.L. and X.T.; funding acquisition, H.L. and X.T. All authors have read and agreed to the published version of the manuscript.

**Funding:** This research was jointly supported by the National Key Research and Development Program of China (2019YFD1100104), National Natural Science Foundation of China (51978198, 52000049), State Key Laboratory of Urban Water Resource and Environment (2020DX04), and China Postdoctoral Science Foundation (2019M651290) and (220T130153).

**Institutional Review Board Statement:** Not applicable.

**Informed Consent Statement:** Not applicable.

**Conflicts of Interest:** The authors declare no conflict of interest.

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
