# Peer review of "Effects of Filtration Mode on the Performance of Gravity-Driven Membrane (GDM) Filtration: Cross-Flow Filtration and Dead-End Filtration"

_water, doi:10.3390/w14020190_

Round 1

Reviewer 1 Report

Review of water-1526445

  1. Please add a benchmarking table having information of literatures of similar gravity-driven membrane filtration, and about the performance (their TMP, rejection, flux, operation conditions, etc.). This table is important as a display of improvement of performance and enhancement of knowledge brought by this submission.
  2. The reduction shown in this manuscript is 55-64%, which is not so special, and therefore must be benchmarked with others references.
  3. Write the units in the form of X Y-1, not X/Y. This is in order to be consistent with the wavenumber having the unit cm-1. Do not write LMH, L/(m2h), mg/L, etc. A messy inconsistency of units can be seen in Table 1. Write consistently the unit for flux as L m-2 h-1 in all figures, tables, and paragraphs.
  4. Change all mg/L to be mg L-1.
  5. For the unit of TMP, it is better to use a consistent unit mbar, and not cm. Moreover, the correct unit is actually cmH2
  6. Cite a recent 2021 paper about gravity driven ultrafiltration: Ishak et al, “Decontaminate river water via portable gravity-driven ultrafiltration (GDU) unit: Fouling and cleaning efficiencies studies”, J Env Chem Eng 9 (2021) 106213 https://doi.org/10.1016/j.jece.2021.106213
  7. Line 30: …remove tyrosine (a fluorescent protein like-substance)..
  8. Line 30: Add the information of tyrosine in the Methods section. Do you use one type of tyrosine or multiple types of tyrosines? If there are multiple types of tyrosines then revise the abstract accordingly.
  9. Line 60: Start the genus of this scientific name with uppercase letter T (Tetrahymena pyriformis).
  10. Line 70: Remove the excessive spaces.
  11. Line 70: …and atrazine). However, the…
  12. Line 140: Do you mean “Chinese National Standard”?
  13. Line 249: 30th day --> with superscripted th
  14. Line 267-268: What are the units for the numbers there?
  15. Line 299-300: What do you mean with “…in CGDM were…slightly higher than CGDM…”. Please correct it.
  16. Line 351: The correct name of a co-author for Reference 14 is L. V. de Souza Santos.

Reviewer 2 Report

This paper should be published because of its high novelty and intererest for researchers working on water filtration issues. I have only some minor commments because the running text, methods used and overall presentation is very good.

  1. Line 60: Latin name must be with capital letter, Tetrahymena
  2. Line 65: A. Chomiak et al. Should be Chomiak et al.
  3. Line 129: Be aware about the use of units used throughout the ms. L/m2/hr  or L m-2 h-1. 
  4. Fig. 3b. Y-axis should finish at 100%

Round 2

Reviewer 1 Report

Review of water-1526445-v2
The authors have addressed all inquiries previously raised. The manuscript can be accepted now.